# Adaptation Strategy of the Planula Strobilation in Moon Jelly, *Aurelia coerulea* to Acidic Environments in Terms of Statolith Formation

**DOI:** 10.3390/ani15131999

**Published:** 2025-07-07

**Authors:** Yuka Maeda, Hiroshi Miyake, Nobuo Suzuki, Shouzo Ogiso

**Affiliations:** 1School of Marine Biosciences, Kitasato University, 1-15-1 Kitazato, Minami-ku, Sagamihara 252-0373, Kanagawa, Japan; 2Noto Marine Laboratory, Institute of Nature and Environmental Technology, Kanazawa University, Mu-4-1 Ogi, Noto-cho 927-0553, Ishikawa, Japan; nobuos@staff.kanazawa-u.ac.jp (N.S.); shozoogiso@se.kanazawa-u.ac.jp (S.O.)

**Keywords:** ocean acidification, life cycle, statolith, strobilation, biological minimum form

## Abstract

Ocean acidification poses a significant threat to marine invertebrates with calcium-based structures. This study investigated the effects of low pH on two types of strobilation in *Aurelia coerulea*: polyp-strobilation (conventional asexual reproduction from polyps) and planula-strobilation (direct development from planulae). Experiments were conducted under pH 6.8, 7.8, and 8.1 conditions to observe morphological changes and statolith formation in ephyrae. Under the pH 6.8 condition, polyp-strobilation failed to produce normal ephyrae, while planula-strobilation succeeded in releasing morphologically normal ephyrae, albeit without statoliths. Under the pH 7.8 condition, both strobilation types produced ephyrae with altered statolith morphology. These statoliths were smaller in size but more numerous than those formed at pH 8.1 as normal pH, suggesting a compensatory mechanism that maintains total statolith mass and potentially preserves function. Planula-strobilated ephyrae had fewer but larger, needle-shaped statoliths, suggesting rapid statolith development. These findings suggest that planula-strobilation functions as a stress-adaptive reproductive strategy, producing the minimum necessary morphology and internal structures to ensure survival in a changing environment. The ability of *Aurelia coerulea* to adjust reproductive strategy and developmental traits under acidified conditions may contribute to its ecological success and persistence under future climate change scenarios.

## 1. Introduction

Ocean acidification caused by increasing CO_2_ emissions has been a major problem on Earth [1]. The Paleozoic and Mesozoic eras are estimated to have experienced ocean acidification because of volcanic activity. The Mg/Ca ratio in ancient sea water samples estimated from fossils of sea urchin spines was below four [2]. In recent years, this ratio has reached 5.2 [2], thus the recent ocean acidification is significantly higher and more serious compared with the previous one. Since the Industrial Revolution, anthropogenic CO_2_ emissions into the atmosphere have increased, resulting in the extensive absorption of this greenhouse gas by the ocean [3,4,5]. Based on representative concentration pathway scenarios (RCP scenarios) regarding greenhouse gas emission levels, the ocean’s pH is predicted to decrease to 7.8 by 2100 [6]. When excess CO_2_ is absorbed by the ocean, it alters the carbonate chemistry and lowers pH, impacting marine life in various ways [1,7,8,9,10]. One of the major significant impacts is the reduction in shell formation, as an increased CO_2_ concentration leads to a decrease in carbonate ions, which are necessary for forming calcium carbonate shells [11,12,13]. Thus, damage has been reported in shell-forming organisms such as planktonic foraminifera and pteropods [14,15], mussels [16], corals [13,17,18], and sea urchins [19]. Beyond these effects on calcification organisms, decreased pH can also disrupt physiological processes by causing acidosis, leading to reduced metabolism, growth, and reproduction rates in marine organisms such as mussels [16]. Studies on the relationship between marine organisms and ocean acidification have tended to focus on calcifying species [20,21,22]. On the other hand, ocean acidification has been found to affect the metabolism, growth, and reproduction rates of organisms, even in the absence of calcification processes [23,24,25,26].

In recent years, jellyfish bloom has been a serious ocean problem. In particular, the moon jelly, *Aurelia,* which appear in oceans worldwide, has frequently been reported to cause jellyfish blooms [27,28]. The cause of jellyfish bloom is derived from the adaptability of *Aurelia* to recent marine environmental changes such as increasing seawater temperatures caused by global warming, eutrophication, and overfishing [29,30]. When *Aurelia* medusae form jellyfish blooms, fishing damage occurs due to the medusae clogging fishing nets as well as the invasion into intake pipes of power plants, leading to shutdowns [27,29]. Furthermore, jellyfish bloom impacts marine ecosystems by increasing the predation pressure on zooplankton and larval fish [31] as well as causing competition for food with zooplanktivorous fish [32,33,34]. Especially in Japan, where jellyfish blooms of *Aurelia* have frequently occurred, the ecology of *Aurelia* in the field has been studied [35,36,37].

The typical life cycle of *Aurelia* alternates between the polyp (sessile asexual reproductive stage) and medusa (planktonic sexual reproductive stage). Polyp-to-medusa transition is caused by strobilation by polyp. Strobilated polyps release ephyrae, the juveniles of medusae. Ephyrae have statoliths as an equilibrium organ [38]. In scyphozoans, statoliths are contained within the statocysts, sac-like balance organs that detect the movement of the statoliths relative to gravity, enabling the animal to determine its orientation within the water column [39,40,41,42]. In addition, statocysts are located at the apical end of the rhopalia, sensory structures that consist of epidermal neurons, chemoreceptor pits, a statocyst, and an ocellus [38]. The rhopalia are located at the margin of the medusa umbrella [40]. The primary component of statoliths is calcium sulfate hemihydrate [41,43]. The statocyst is also composed of endodermal cells, lithocytes, which take up calcium and sulfate from seawater and form the statoliths [40,44]. Ocean acidification increases the concentration of hydrogen ions in seawater. Hydrogen ions bind with sulfate ions, reducing the concentration of sulfate ions available to bond with calcium ions [3,23]. Thus, statolith formation may be inhibited in acidic environments. Therefore, the impact of ocean acidification on *Aurelia* has been studied in relation to the statoliths of ephyrae [45]. Winans and Purcell (2010) reared polyps of *Aurelia labiata* under acidic conditions and observed ephyrae released from the polyps in these conditions [45]. They reported that the size of the statoliths was reduced when ephyrae were kept in acidic conditions. However, the effects of smaller statoliths on the adaptive capacity of ephyrae remain unknown. Furthermore, the pulsing rate of *Aurelia* ephyrae decreased when exposed to low pH conditions [46]. These findings suggest that survival in acidic oceans may be difficult for *Aurelia* medusae.

Statolith formation occurs during ephyra development [47,48,49]. In *Aurelia*, there are two life cycles with different ephyra formation periods. *A. aurita* in Germany and *A. aurita* (later revised to “*A. coerulea*” [50]) in Japan (Figure 1) omit the polyp stage after the planula settles on a substrate and directly metamorphoses into an ephyra [36,37,51,52,53,54,55,56].

This type of development is known as “direct development” [36,40,52,53,54,57]. Typically, ephyra formation by polyp strobilation takes about one month and results in a poly-disc (forming several ephyrae), whereas ephyra formation by direct development takes only about a week and results in a mono-disc (forming only a ephyra) [36,53] (Figure 2).

Normal-sized planulae are approximately 200–300 µm in length, whereas planulae that directly metamorphose into ephyrae are larger, ranging from 500 to 800 µm [37,53]. These large planulae can develop into ephyrae at low water temperatures (e.g., 11 °C); however, at higher water temperatures (>20 °C), they cannot develop into ephyrae but into polyps [55]. Strobilation, which is ephyra formation by polyps, also progresses at low water temperatures [53]. Therefore, the direct development of *A. coerulea* is called “planula-strobilation” [58]. The speed of ephyra formation via planula-strobilation is approximately four times faster than that of conventional polyp-strobilation [36,53]. Ephyrae formed through planula-strobilation (i.e., through rapid ephyra formation) may be more susceptible to an acidic environment due to their simplified morphology. Therefore, they represent an excellent material for observing the effects of acidic environments on both ephyrae and statolith formation through direct exposure.

In this study, we hypothesized that the ephyrae produced by planula-strobilation would be more affected by the acidic environment than those formed via polyp-strobilation. To test this, this study aimed to examine ephyra formation in both the polyp-strobilation and planula-strobilation of *A. coerulea* under low pH conditions.

## 2. Materials and Methods

### 2.1. Experimental System Setup

Three experimental tanks (width 18 cm × diameter 31 cm × height 24 cm; Five Plan, GEX, Osaka, Japan) were each filled with 10.5 L of 33‰ filtered seawater (1 µm) and gently aerated using an air pump (AquaCore S size, SUISAKU, Taito, Japan) and a filter (Roka boy S size, GEX, Osaka, Japan). Each experimental tank was regulated using a pH controller (Fukurow, AQUA GEEK, Kawaguchi, Japan) and maintained at one of three pH levels: 6.8, 7.8, and 8.1. A single tank was used for each pH condition. The pH 6.8 condition reflects the CO_2_ seep area in Kagoshima Bay, where *A. coerulea* polyps have been observed in the field [59], and is the lowest record pH at which *A. coerulea* polyps naturally occur. The pH 7.8 condition represents the predicted average ocean pH in the year 2100 based on the RCP scenario [6]. When the pH in the tank exceeds the target value, the pH controller activates a solenoid valve to inject CO_2_ and restore the target pH value. All three experimental tanks were placed in a large tank (width 90 cm × diameter 60 cm × height 45 cm), and the temperature was maintained using a water chiller (Aqua cooler 40-II, Marukan, Osaka, Japan).

### 2.2. Obtaining Polyp-Strobilated Ephyrae

In April 2022, a female medusa of *A. coerulea* with a bell diameter of approximately 20 cm was collected at Ushitsu Port, Ishikawa Prefecture (Figure 1). Planulae with a size of 200–300 µm in the long axis were collected from the medusa. These planulae were placed in a 250 mL bottle (i-boy, AS ONE, Osaka, Japan) filled with local seawater and transported to the laboratory. These were then transferred to a plastic dish (diameter 10 cm and height 5 cm) and developed into polyps.

From the resulting polyps, one individual was transferred to a Petri dish (diameter 8.5 cm and height 2 cm) to serve as the “mother polyp”. This polyp was allowed to settle near the center of the Petri dish. A total of nine Petri dishes, each containing one polyp settled at the center, were prepared. Three of these Petri dishes were placed in each experimental tank. The temperature of the experimental tanks was maintained at 23 °C. To consider only the effect of pH, the first budded polyp from the mother polyp in each Petri dish was left as the “initial polyp”, and the mother polyp was removed. The polyps were then cultured for two months to form polyp colonies derived from the initial polyps. These colonies were fed *Artemia* nauplii daily, and water changes were conducted once a week.

Polyps strobilate when the water temperature decreases [53,60]. Therefore, the water temperature was also decreased from 23 °C to 11 °C to induce strobilation in this study. The Petri dishes containing the polyp colonies were placed into fish breeder nets (S-5330, Sudo, Nagoya, Japan) installed within each experimental tank. Forty-five days after decreasing the water temperature, the polyps formed strobilae. Newly released (0-day-old) ephyrae were collected daily from the breeder net. Under the pH 7.8 and pH 8.1 conditions, a total of 7 and 10 polyp-strobilated ephyrae were randomly selected for the observation of body morphology and statolith formation. Each ephyra was measured once; thus, the sample size (*n* = 7 or 10) refers to individual organisms derived from a single tank per pH condition. While technical replicates were not included, each data point represents a distinct individual, providing variation within the tank.

### 2.3. Obtaining Planula-Strobilated Ephyrae

In June 2023, three female medusae of *A. coerulea*, each with a diameter of 15.5–20.0 cm, were collected in Tsuruga Bay, Fukui Prefecture (Figure 1). Planulae ranging from 375 to 823 μm (mean 550 ± 85.4 μm, *n* = 50) in the long axis were collected from these medusae. All collected planulae were divided into three groups, and each group was placed in 500 mL bottles (Wide-mouth T-type bottle, AS ONE, Osaka, Japan) filled with seawater adjusted to the target pH conditions in advance. The planulae were transported to the laboratory and transferred to a 100 mL bottle (Clear wide-mouth bottle, AS ONE) with a 100 µm mesh lid. These bottles were then placed in experimental tanks maintained at 11 °C to induce planula-strobilation. Newly released (0-day-old) ephyrae were collected daily from each bottle. Ten ephyrae were randomly selected for the observation of body morphology and statolith formation. Each ephyra was measured once; thus, the sample size (*n* = 10) refers to individual organisms derived from a single tank per pH condition. While technical replicates were not included, each data point represents a distinct individual, providing variation within the tank.

### 2.4. Observation of the Body Morphology of Ephyra

Ephyrae with a complete marginal lappet (eight marginal lappet, one pair of rhopalial lappet on the tip of each marginal lappet) were used for morphological observation. The ephyrae were examined under a stereomicroscope (SZX16, Olympus, Hachiōji, Japan) and photographed using the imaging software, cellSens standard (ver. 3.2), Olympus, Hachiōji, Japan). Thereafter, the total body diameter (TBD, refer to Figure 3) was measured as the body size of the ephyrae using the imaging software, ImageJ (ver. 1.53k) [61]. The ratio of central disc diameter (CDD, refer to Figure 3) to TBD as CDD/TBD × 100 (%) was also calculated from the photograph. This ratio represents a characteristic body proportion of planula-strobilated ephyrae [58].

### 2.5. Observation of Statoliths

Following the morphological observations, the ephyra was used for the observation of statoliths. In accordance with the study of Winans and Purcell (2010), a small amount of glycerin was dropped onto an ephyra placed on a glass slide (S1226, Matsunami, Osaka, Japan) [45]. Afterward, a small amount of 90% ethanol was mixed with the glycerin to reduce the viscosity. A cover glass was gently placed over the specimen on the slide and then lightly pressed and rotated with a finger to break the statocyst and expose the statoliths. According to Winans and Purcell (2010), three statocysts were selected randomly and examined for each ephyra [45].

Statoliths were observed using a biological microscope (Labophoto, Nikon, Shinagawa, Japan), and photo images were captured using a digital camera (EOS Kiss X6i, Canon, Ōta, Japan) mounted on the microscope. The long (l) and short (w) axis of all statoliths were measured from the photo images using ImageJ (Figure 4). The number of statoliths per statocyst was also counted. Broken statoliths were included in the count but excluded from the axis measurements. The statolith size was calculated as the product of the long axis and short axis. In addition, the aspect ratio was defined as the long axis divided by the short axis (l/w). For each statocyst, the average size and aspect ratio of the statolith were calculated.

### 2.6. Statistical Analysis

To assess whether significant differences existed in the ephyra morphology and statolith size across different pH conditions and strobilation types, we performed statistical analyses using R version 4.3.1 [62]. Levene’s test was applied to examine the assumption of the homogeneity of variances. For datasets meeting this assumption, we conducted a traditional two-way analysis of variance (two-way ANOVA). When the assumption was violated, we employed a robust two-way ANOVA using Yuen’s trimmed means, which is less sensitive to heteroscedasticity and outliers. This approach was chosen because it provides more reliable results when data violate the homogeneity of variance assumption and contain potential outliers, which was the case for some of our datasets. In particular, the trimmed means reduce the influence of extreme values, which were observed in several morphological measurements. This analysis was performed using the ‘t2way’ function in the WRS2 package in R [63]. Subsequently, to identify which specific group pairs showed significant differences, we performed robust post hoc comparisons using the Lincon function from the same package. However, under the pH 6.8 condition, the polyp-strobilated ephyrae were deformed, while the planula-strobilated ephyrae were morphologically normal. Therefore, to analyze the body morphology of planula-strobilated ephyrae across the three pH conditions, including pH 6.8, we first confirmed the homogeneity of variances using Bartlett’s test, and then conducted a one-way ANOVA. Afterward, significant differences between each pH condition were confirmed by Tukey’s HSD.

## 3. Results

### 3.1. Luvene’s Test

For the body morphology of the ephyrae, three parameters were analyzed: TBD, CDD, and CDD/TBD ratio. Levene’s test confirmed the homogeneity of variances for each parameter (*p* = 0.082, 0.198, and 0.151, respectively), therefore, a traditional two-way ANOVA was conducted for each.

For statoliths, the measured parameters were number, size, and aspect ratio. Levene’s test indicated a heterogeneity of variances in all cases (*p* = 0.004, <0.001, and <0.001, respectively), and thus a robust two-way ANOVA was applied accordingly.

### 3.2. Body Morphology of the Ephyrae

#### 3.2.1. Results of Traditional Two-Way ANOVA

Traditional two-way ANOVAs and independent *t*-tests were conducted to examine the effects of strobilation type (polyp-strobilation vs. planula-strobilation) and pH on the TBD, CDD, and the CDD/TBD ratio. Under the pH 8.1 conditions, the strobilation type had a significant effect on all three variables: TBD (*F* = 339.07, *p* < 0.001), CDD (*F* = 261.21, *p* < 0.001), and CDD/TBD ratio (*F* = 38.53, *p* < 0.001). Under the pH 7.8 conditions, the strobilation type also significantly affected all variables: TBD (*F* = 35.12, *p* < 0.001), CDD (*F* = 32.80, *p* < 0.001), and CDD/TBD ratio (*F* = 21.80, *p* < 0.001).

When examining the effect of pH within polyp-strobilation, no significant difference was found in TBD (*F* = 0.28, *p* = 0.61) or CDD (*F* = 1.42, *p* = 0.27), while a significant effect was observed in the CDD/TBD ratio (*F* = 11.30, *p* = 0.004).

#### 3.2.2. Polyp-Strobilated Ephyrae

Under the pH 6.8 condition, all polyps failed to develop into strobilae. Even though the polyps developed into strobilae, the strobilae were abnormal. No strobilae produced ephyrae with normal marginal lappets. Moreover, these malformed strobilae lacked a rhopalial lappet and ultimately died (Figure 5). Under the pH 7.8 condition, the TBD, CDD and CDD/TBD were 2.98 ± 0.44 mm, 1.18 ± 0.20 mm, and 39.3 ± 1.0%, respectively (Table 1, Figure 5). Under the pH 8.1 condition, the TBD, CDD and CDD/TBD ratio was 3.08 ± 0.14 mm, 1.28 ± 0.09 mm, and 41.5 ± 1.3%, respectively (Table 1, Figure 5).

#### 3.2.3. Planula-Strobilated Ephyrae

Under the pH 6.8 condition, the TBD, CDD, and CDD/TBD ratio were 1.37 ± 0.17 mm, 0.62 ± 0.07 mm, and 45.4 ± 4.8%, respectively (Table 1, Figure 5). Under the pH 7.8 condition, the TBD, CDD, and CDD/TBD ratio were 1.95 ± 0.22 mm, 0.68 ± 0.09 mm, and 35.0 ± 2.5%, respectively (Table 1, Figure 5). Under the pH 8.1 condition, the TBD, CDD, and CDD/TBD ratio were 1.72 ± 0.17 mm, 0.61 ± 0.09 mm, and 35.3 ± 2.6%, respectively (Table 1, Figure 5).

For body morphology, three parameters were analyzed: the TBD, CDD, and CDD/TBD ratio. Bartlett’s test confirmed the homogeneity of variances for each parameter (*p* = 0.647, 0.817, and 0.067, respectively); therefore, a one-way ANOVA was conducted for each.

For TBD, the one-way ANOVA revealed a significant effect of pH (*F* = 18.552, *p* < 0.001). For CDD, no significant effect of pH was found (*F* = 0.116, *p* = 0.736). For the CDD/TBD ratio, a significant effect of pH was observed (*F* = 42.545, *p* < 0.001).

Subsequently, Tukey’s HSD test was performed to examine significant differences between each pH condition. In the TBD group, significant differences were detected among all pH groups: pH 6.8 vs. 7.8 (*p* < 0.001), pH 6.8 vs. 8.1 (*p* = 0.002), and pH 7.8 vs. 8.1 (*p* = 0.04). In contrast, no significant differences were found among the pH groups in the CDD group (*p* > 0.05). Regarding the CDD/TBD ratio, significant differences were detected between pH 6.8 and the other two groups (*p* < 0.001), while no significant difference was found between pH 7.8 and 8.1 (*p* = 0.86) (Table 2).

### 3.3. Statoliths of the Ephyrae

#### 3.3.1. Results of Yuen’s Trimmed Means Two-Way ANOVA

For the average number of statoliths per statocyst, a robust two-way ANOVA using Yuen’s trimmed means revealed significant main effects of strobilation type (value = 89.50, *p* = 0.001) and pH condition (value = 7.51, *p* = 0.012), but no significant interaction effect (value = 0.07, *p* = 0.799).

In the statolith size, a robust two-way ANOVA using Yuen’s trimmed means revealed significant main effects of strobilation type (value = 107.63, *p* = 0.001) and pH condition (value = 6.59, *p* = 0.015), but no significant interaction effect (value = 0.26, *p* = 0.612).

In the aspect ratio, a robust two-way ANOVA using Yuen’s trimmed means revealed significant main effects of strobilation type (value = 278.00, *p* = 0.001), pH condition (value = 11.53, *p* = 0.002), and a significant interaction effect (value = 5.84, *p* = 0.021).

#### 3.3.2. Polyp-Strobilated Ephyrae

Under the pH 6.8 condition, the deformed polyp-strobilated ephyrae lacked rhopalial lappets, and as a result, did not possess statocysts—and consequently, no statoliths. Under the pH 7.8 condition, the average number of statoliths per statocyst was 28.6 ± 6.28. The mean statolith size was 66.2 ± 10.3 μm^2^ and the mean aspect ratio (long axis/short axis) was 1.5 ± 0.18 (Table 3, Figure 6). Under the pH 8.1 condition, the average number of statoliths per statocyst was 24.6 ± 3.22. The mean statolith size was 79.1 ± 7.73 μm^2^ and the aspect ratio was 2.1 ± 0.28 (Table 3, Figure 6).

#### 3.3.3. Planula-Strobilated Ephyrae

Under pH 6.8, all planula-strobilated ephyrae were observed to possess statocysts, but none contained statoliths (Figure 7). Although the ephyrae were compressed and rotated under the cover glass to break the statocysts and expose statoliths, no statoliths could be found. Under the pH 7.8 condition, the average number of statoliths per statocyst was 17.9 ± 3.53. The mean statolith size and the aspect ratio were 111.0 ± 20.4 μm^2^ and 3.5 ± 0.66, respectively (Table 3, Figure 6). Under the pH 8.1 condition, the average number of statoliths per statocyst was 15.4 ± 3.35. The mean statolith size and the aspect ratio were 114.6 ± 31.8 μm^2^ and 3.6 ± 0.72, respectively (Table 3, Figure 6).

## 4. Discussion

Ephyrae with abnormal marginal lappets were produced by polyp-strobilation under the pH 6.8 condition. In contrast, planula-strobilated ephyrae could form normal marginal lappets and they swam normally, even at pH 6.8. This difference may be attributed to the duration required for ephyra formation in the two strobilation types. The period of ephyra formation by polyp-strobilation requires approximately one month, whereas planula-strobilation can produce ephyrae within just one week [36,53]. Because the exposure to the low-pH environment was shorter in planula-strobilation, normal-shaped ephyrae could be produced, even under pH 6.8. However, polyps of *Sanderia malayensis* and *Aurelia coerulea* (described as *Aurelia aurita*) have been found on tubes of vestimentiferan tubeworms at depths of an 80–110 m area, where the pH was 7.0–7.7 by the CO_2_ seep environment in Kagoshima Bay, Japan [59]. Strobilae of *S. malayensis* were found in situ. These polyps have adapted to the low-pH environment. In our study, polyps were reared for approximately 110 days (two months and 45 days) under low-pH seawater. This experimental duration may have been too short for the polyps of *A. coerulea* to adapt to the acidic environment at pH 6.8. If long-term experiments are conducted, normal strobilation might occur even at pH 6.8, as observed in Kagoshima Bay.

*Aurelia* ephyrae can survive short-term under exposure to high-*p*CO_2_ conditions [64]. In the present study, although planula-strobilated ephyrae had normal marginal lappets, statoliths were absent at pH 6.8 (Figure 7). *A. aurita* polyps reared in sulfate-free artificial seawater failed to form statoliths during strobilation and released ephyrae exhibiting swimming abnormalities [48]. Therefore, even ephyra produced by planula-strobilation may have difficulty swimming at pH 6.8.

In polyp-strobilation, the body size of ephyrae under the pH 7.8 condition was smaller than that under the pH 8.1 condition. The shrinking of ephyra bodies also occurred when *A. aurita* ephyrae were maintained at pH 7.6 for 7 days [46]. However, in the present study, no significant difference in the body size of polyp-strobilated ephyrae was observed between the pH 7.8 and pH 8.1 conditions (Table 1, Figure 8). Moreover, the body size of planula-strobilated ephyrae did not shrink even under the pH 7.8 condition. This suggests that an acidic environment such as pH 7.8, which is predicted to decrease by the year 2100, does not affect the growth of the ephyra body. On the other hand, the CDD/TBD ratio of polyp-strobilated ephyrae was lower under the pH 7.8 condition than under the pH 8.1 condition. A lower CDD/TBD ratio indicates that the marginal lappet length is relatively longer compared with the CDD. The marginal lappet generates propulsion to allow the ephyrae to swim [65,66]. Therefore, polyp-strobilated ephyrae may have a better swimming ability at pH 7.8 than at pH 8.1.

In both strobilation types, the statoliths were smaller at pH 7.8 than at the control (pH 8.1). The size of the statoliths of *A. labiata* ephyrae reared in an acidic environment decreased, but the effect of smaller statoliths on the ephyra remains unclear [45]. However, in both strobilation types, the number of statoliths per statocyst increased at pH 7.8 compared with the control (pH 8.1).

Calcium sulfate hemihydrate is the primary component of statoliths [41]. The growth rate of the crystal nuclei of calcium sulfate hemihydrate is slowed down under high-pH conditions in the pH range of 1.2–8.0 [67]. When the growth rate of crystal nuclei is reduced, fewer crystals are formed. Thus, the number of statoliths per statocyst was likely to be lower under the pH 8.1 condition than under the pH 7.8 condition in this study. The weight of the statoliths enables the jellyfish’s sense of balance. Considering the total number of statoliths, it is suggested that despite the smaller size of the statoliths at pH 7.8, the increased number of statoliths at this pH may maintain the total statolith mass required for normal swimming. It has also been reported that the ephyrae of *A. aurita* were robust to acidic conditions because of their normal swimming activity and low mortality under acidic conditions [68]. Consistent with this, our results showed that polyp-strobilation proceeded normally and yielded morphologically complete ephyrae at pH 7.8, suggesting that ephyrae can be formed and survive under acidic environments such as at pH 7.8.

Planula-strobilated ephyrae have larger statoliths than polyp-strobilated ephyrae. This is thought to be due to the lower number of statoliths per statocyst in planula-strobilated ephyrae, and may allow more material to be allocated to each individual crystal. Consequently, the planula-strobilated ephyrae possessed fewer but larger statoliths compared with the polyp-strobilated ephyrae. Furthermore, when the number and size of the statoliths were multiplied, the conceptual mass of the statoliths was not significantly different between the two strobilation types under either pH condition (Figure 9). This result indicates that despite the rapid formation of planula-strobilated ephyrae, they can still form a total number of statoliths comparable to the polyp-strobilated ephyrae.

The planula-strobilated ephyrae had needle-shaped statoliths with higher aspect ratios than those derived from polyp-strobilation. Calcium sulfate hemihydrate forms needle-like crystals at the initial stage of formation, which increases in both length and thickness as it grows [69]. Thus, the planula-strobilated ephyrae had the characteristics of statoliths in the early stage of crystal formation. Based on previous reports, ephyrae are released 10 days after the formation of a rhopalium during polyp-strobilation [47]. In contrast, observations in this study revealed that planula-strobilation released ephyrae just 0–3.5 days after the formation of the rhopalium. *Sanderia malayensis* undergoes mono-disc type strobilation within only 2–4 days [70]. This also suggests that the statolith formation period is also short. In fact, needle-shaped statoliths were observed in the newly released ephyrae of *S. malayensis*. Since needle-shaped statoliths are not found in moon jellyfish (*A. aurita*), they were previously considered species-specific [70]. However, our findings suggested that needle-shaped statoliths may indicate the early stages of statolith formation.

The ephyrae of *A. aurita* (later reclassified as *A. coerulea*) with minimum body size are able to survive after recovering from starvation-induced damage [71]. This minimum size is almost the same as the body size of planula-strobilated ephyrae [58]. Therefore, the size of planula-strobilated ephyrae is thought to be the biological minimum size for survival in nature [58]. Moreover, planula-strobilation refers to the short-period formation of ephyrae. Planula-strobilation forms minimal-state statoliths that are necessary for survival and release the ephyrae shortly. Thus, planula-strobilated ephyrae represent the biological minimum state, not only in terms of external morphology, but also in terms of internal statolith characteristics. Body proportions can be used to distinguish between planula-strobilated and polyp-strobilated ephyrae [58]. This study indicates that the morphology of the statolith can also be used as an indicator for distinguishing between planula-strobilated and polyp-strobilated ephyrae.

Regarding significant differences between pH 7.8 and pH 8.1 for each strobilation type, the number of items with a significant difference was fewer for planula-strobilation than for polyp-strobilation (Table 3, Figure 10). In this study, we hypothesized that the planula-strobilated ephyrae would be more affected by the acidic environment than polyp-strobilated ephyrae in the Introduction. On the contrary, planula-strobilation was less susceptible to acidification damage than polyp-strobilation. Because the duration of ephyra formation in planula-strobilation is shorter than in polyp-strobilation, the 0-day-old ephyrae derived from planula-strobilation may have experienced less exposure to acidic conditions, resulting in reducing the number of items with a significant difference.

If atmospheric CO_2_ continues to rise in the future, then global warming will intensify. For example, in scyphozoan *Rhizostoma pulmo* ephyrae, their statoliths become smaller under acidic conditions, while under warmer conditions, they grow larger, and when both conditions are combined, the statoliths were still synthesized to be slightly larger [72]. Thus, *A. coerulea* ephyrae may also form slightly larger statoliths under acidic, warm conditions.

In general, polyp-strobilation in *Aurelia* species is dependent on low temperatures [60,73]. Planula-strobilation also occurs depending on the low water temperature [55]. In the strobilation experiment assuming a rise in the winter minimum water temperature from the current 5 °C to 10 °C under warming scenarios, the strobilation rate of *A. aurita* increased, the duration of strobilation was extended, and the number of ephyrae produced per polyp also increased [74]. Similarly, strobilation rates in *A. labiata* increased at a higher prestrobilation temperature [75]. From the above, global warming may accelerate strobilation in *Aurelia*. Thus, polyps can normally strobilate even under acidic conditions such as pH 7.8, which is predicted to decrease by 2100. *Aurelia* could adapt to global warming and ocean acidification. Planula-strobilation also forms ephyrae more quickly at higher water temperatures [37]. In addition, even during periods of low food availability, *A. coerulea* can enhance its survival rate through planula-strobilation [58]. Furthermore, in planula-strobilation, the body morphology of the ephyra and the formation of statoliths are less affected by acidification than polyp-strobilation. Based on our findings, the planula-strobilation type represents an adaptive strategy to rapidly produce viable ephyrae under challenging environmental conditions by minimizing body and statolith formation to the essential levels necessary for survival.

## 5. Conclusions

This study revealed that ocean acidification affects strobilation and statolith formation in *Aurelia coerulea*, with significant differences observed between polyp-strobilation and planula-strobilation. Under the severely acidic condition of pH 6.8, polyp-strobilation was inhibited, while planula-strobilation could still produce morphologically normal ephyrae, although no statolith formation was observed. At pH 7.8, both strobilation types formed ephyrae with altered statolith characteristics—smaller but more numerous crystals—likely maintaining the total statolith mass necessary for balance and movement. The needle-shaped statoliths observed in planula-strobilated ephyrae suggest early-stage crystal formation due to rapid development. These findings indicate that planula-strobilation serves as an adaptive reproductive strategy under environmental stress, enabling survival in suboptimal conditions. Additionally, the body proportions and statolith morphology of ephyrae may serve as reliable indicators to differentiate between strobilation types. Given the predicted decline in ocean pH by 2100, this study suggests that *A. coerulea* may maintain reproductive and developmental success in future ocean conditions through flexible strobilation mechanisms and durable physiological adaptability.

In future research, it would be valuable to apply different methods on the ephyrae of *A. coerulea*, such as rearing under high temperatures, low dissolved oxygen levels, and acidic conditions, to simulate the multifunctorial stressors predicted in future ocean environments. Moreover, expanding such studies to other jellyfish species, such as the scyohozoans *Chrysaora*, *Rhopilema*, and *Nemopilema* as well as the cubozoan *Carybdea*, and to marine invertebrates that utilize calcium carbonate in their physiological processes, such as pteropods, mollusks, and echinoderms, could provide broader insights into organismal resilience to ocean acidification. These multifactorial experiments may clarify how combinations of environmental stressors affect key developmental processes such as strobilation, statolith formation, and polyp performance. Additionally, transcriptomic or proteomic analyses targeting genes involved in calcium transport and biomineralization could help uncover the molecular mechanisms underlying compensatory responses observed under such conditions. In summary, such investigations would enhance our understanding of ecological and evolutionary responses of marine organisms in rapidly changing marine conditions.

## Figures and Tables

**Figure 1 animals-15-01999-f001:**
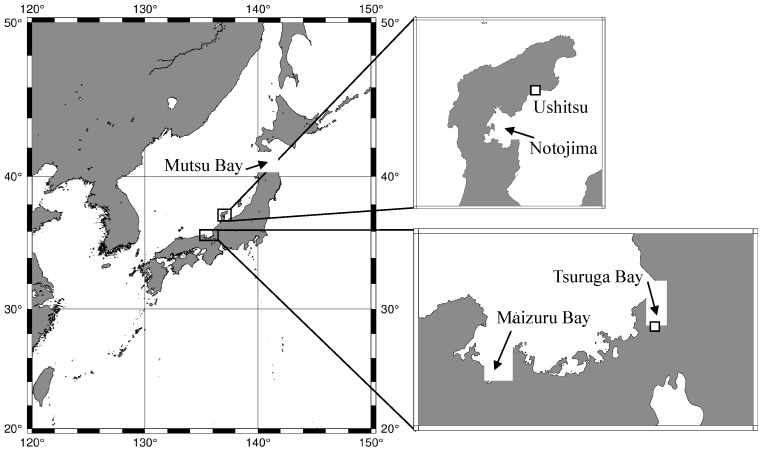
Map of the sampling point (white square) and distribution of the planula-strobilation type *Aurelia* in Japan.

**Figure 2 animals-15-01999-f002:**
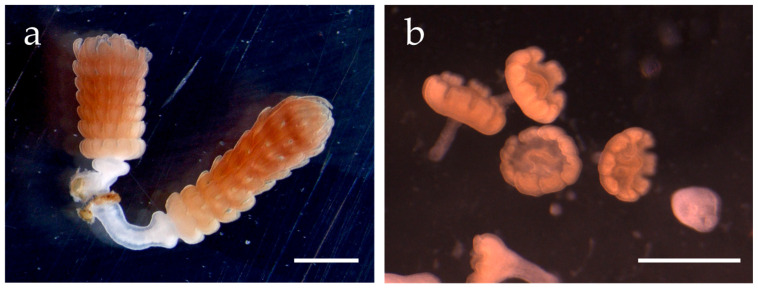
Two types of strobilation in *Aurelia*. (**a**) Polyp-strobilation (poly-disc). (**b**) Planula-strobilation (mono-disc). Scale bar = 1 mm.

**Figure 3 animals-15-01999-f003:**
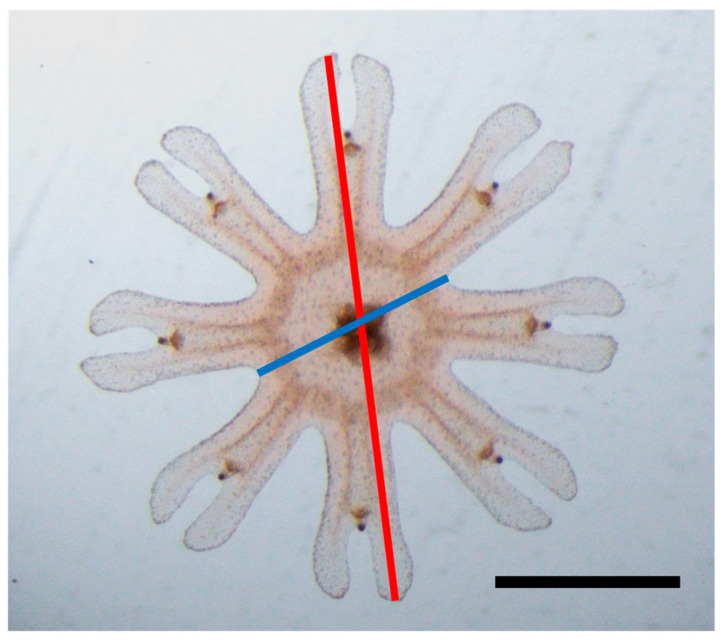
Measuring parts of the ephyra. Red line is the total body diameter (TBD) and the blue line is the central disc diameter (CDD). Scale bar = 1 mm.

**Figure 4 animals-15-01999-f004:**
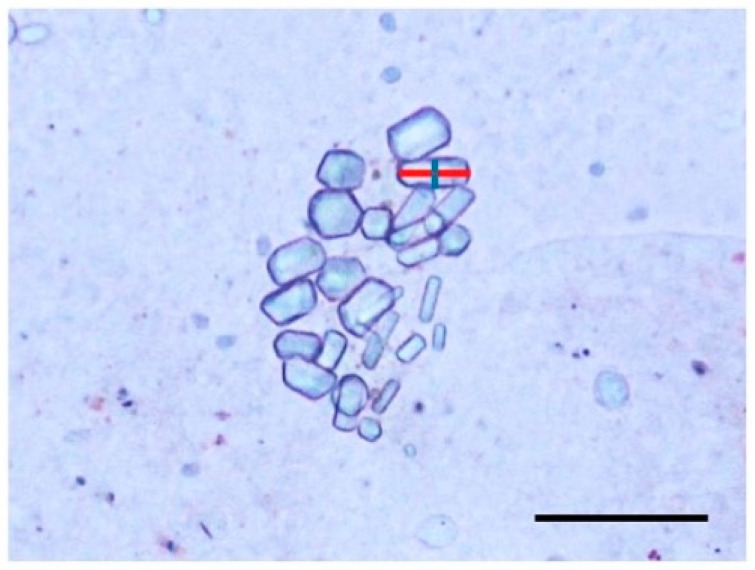
Measuring parts of the statoliths. Red line is the long axis (l) and the blue line is the short axis (w). Scale bar = 50 µm.

**Figure 5 animals-15-01999-f005:**
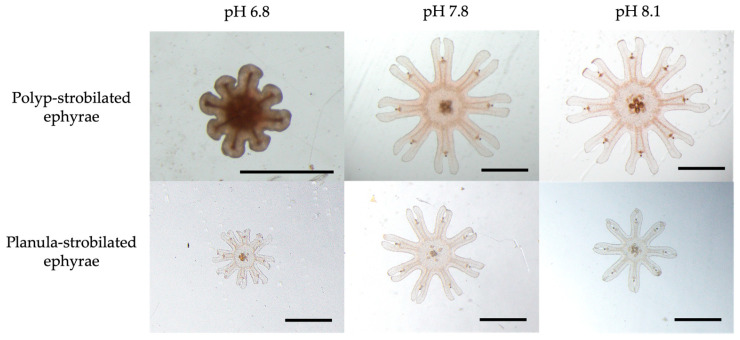
Morphological differences in ephyrae under different pH conditions and strobilation types. Scale bar = 1 mm.

**Figure 6 animals-15-01999-f006:**
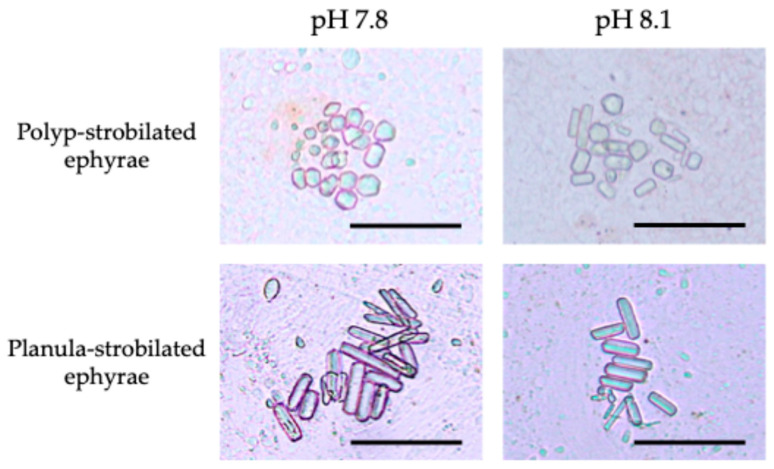
Morphological differences in statoliths between two strobilation types under different pH conditions. Scale bar = 50 μm.

**Figure 7 animals-15-01999-f007:**
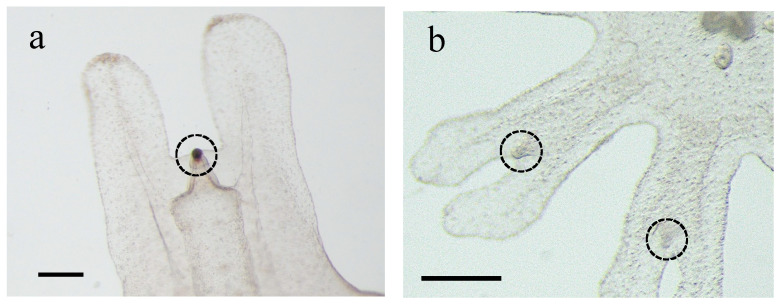
Differences in rhopalia under different acidic conditions. Scale bar = 200 μm. (**a**) Normally formed statoliths in the statocyst under pH 8.1 in polyp-strobilated ephyrae. (**b**) Planula-strobilated ephyrae produced under pH 6.8 had no statoliths in any statocyst.

**Figure 8 animals-15-01999-f008:**
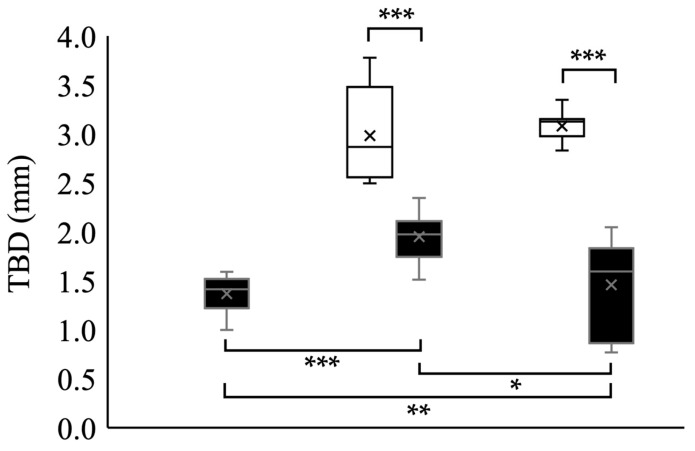
Morphological difference in ephyra body shape. The top and bottom whiskers indicate the maximum and minimum values, respectively. The box edges represent the first and third quartiles, with the line inside showing the median. The cross mark represents the mean. Asterisk shows significant differences: *: *p* < 0.05, **: *p* < 0.01, ***: *p* < 0.001.

**Figure 9 animals-15-01999-f009:**
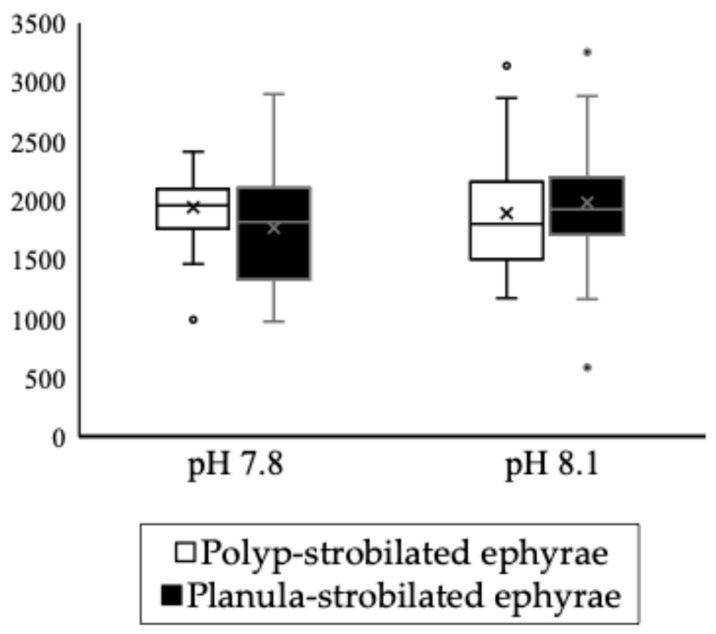
Statolith conceptual mass (statolith size × number per statocyst). The top and bottom whiskers indicate the maximum and minimum values, respectively. The box edges represent the first and third quartiles, with the line inside showing the median. The cross mark represents the mean. Under either pH condition. The dots plotted outside the boxplot represent outliers. There were no significant differences between strobilation types.

**Figure 10 animals-15-01999-f010:**
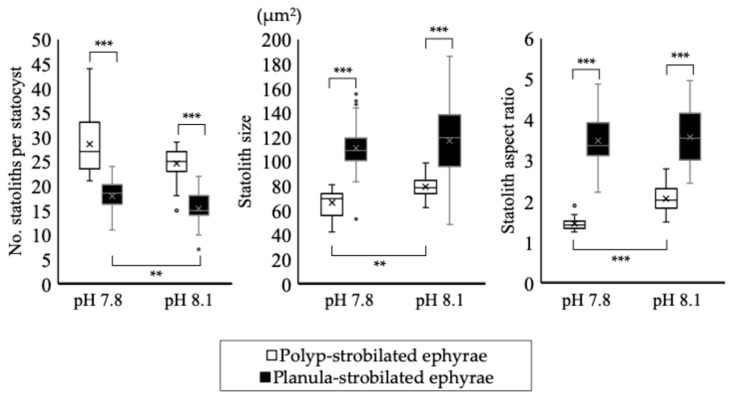
Difference in the morphology of the statoliths. The top and bottom whiskers indicate the maximum and minimum values, respectively. The box edges represent the first and third quartiles, with the line inside showing the median. The dots plotted outside the boxplot represent outliers. The cross mark represents the mean. Asterisk shows significant differences: **: *p* < 0.01, ***: *p* < 0.001.

**Table 1 animals-15-01999-t001:** Effects of pH and strobilation type on ephyra body morphology. Measurements are presented as means (±SE).

	Strobilation Type		Effect	Test Statistic	*p* Value	
	Polyp	Planula				
TBD (mm)						
pH 6.8	-	1.37 (±0.17)	strobilation type at pH 7.8	*F* = 35.12	*p* < 0.001	***
pH 7.8	2.98 (±0.44)	1.95 (±0.22)	strobilation type at pH 8.1	*F* = 339.07	*p* < 0.001	***
pH 8.1	3.08 (±0.14)	1.72 (±0.17)	pH in polyp-strobilation	*F* = 0.28	*p* = 0.61	
			pH in planula-strobilation	Refer to Table 3.		
CDD (mm)						
pH 6.8	-	0.62 (±0.07)	strobilation type at pH 7.8			
pH 7.8	1.18 (±0.20)	0.68 (±0.09)	strobilation type at pH 8.1	*F* = 32.80	*p* < 0.001	***
pH 8.1	1.28 (±0.09)	0.61 (±0.09)	pH in polyp-strobilation	*F* = 261.21	*p* < 0.001	***
			pH in planula-strobilation	Refer to Table 3.		
CDD/TBD (%)						
pH 6.8	-	45.4 (±4.8)	strobilation type at pH 7.8	*F* = 21.80	*p* < 0.001	***
pH 7.8	39.3 (±1.0)	35.0 (±2.5)	strobilation type at pH 8.1	*F* = 38.53	*p* < 0.001	***
pH 8.1	41.5 (±1.3)	35.3 (±2.6)	pH in polyp-strobilation	*F* = 11.30	*p* = 0.004	**
			pH in planula-strobilation	Refer to Table 3.		

The TBD means the total body diameter. The CDD means the central disc diameter. The ratio of CDD to TBD as CDD/TBD × 100 (%) represents a characteristic body proportion of planula-strobilated ephyra. Asterisk shows significant differences: **: *p* < 0.01, ***: *p* < 0.001.

**Table 2 animals-15-01999-t002:** Significant differences between pH in the body morphology of planula-strobilated ephyrae.

TBD (mm)	*p* Value		CDD (mm)	*p* Value	CDD/TBD (%)	*p* Value	
pH 6.8−pH 7.8	*p* < 0.001	***	pH 6.8−pH 7.8	*p* = 0.26	pH 6.8−pH 7.8	*p* < 0.001	***
pH 6.8−pH 8.1	*p* = 0.002	**	pH 6.8−pH 8.1	*p* = 0.97	pH 6.8−pH 8.1	*p* < 0.001	***
pH 7.8-pH 8.1	*p* = 0.04	*	pH 7.8−pH 8.1	*p* = 0.17	pH 7.8−pH 8.1	*p* = 0.86	

The TBD means the total body diameter. The CDD means the central disc diameter. The ratio of CDD to TBD as CDD/TBD × 100 (%) represents a characteristic body proportion of planula-strobilated ephyra. Asterisk shows significant differences: *: *p* < 0.05, **: *p* < 0.01, ***: *p* < 0.001.

**Table 3 animals-15-01999-t003:** Effects of pH and strobilation type on the statolith size, number, and aspect ratio in ephyrae. Values are presented as means (±SE).

	Strobilation Type				
	Polyp	Planula	Effect	*p* Value	
No. ephyrae analyzed					
pH 7.8	7	10		NA	
pH 8.1	10	10		NA	

No. statoliths analyzed					
pH 7.8	600	536		NA	
pH 8.1	737	461		NA	

Mean no. statoliths per statocyst					
pH 7.8	28.6 (±6.28)	17.9 (±3.53)	strobilation type at pH 7.8	*p* < 0.001	***
pH 8.1	24.6 (±3.22)	15.4 (±3.35)	strobilation type at pH 8.1	*p* < 0.001	***
			pH in polyp-strobilation	*p* = 0.178	
			pH in planula-strobilation	*p* = 0.009	**
Mean statolith size (μm^2^)					
pH 7.8	66.2 (±10.3)	111.0 (±20.4)	strobilation type at pH 7.8	*p* < 0.001	***
pH 8.1	79.1 (±7.73)	114.6 (±31.8)	strobilation type at pH 8.1	*p* < 0.001	***
			pH in polyp-strobilation	*p* = 0.003	**
			pH in planula-strobilation	*p* = 0.269	
Mean statolith aspect ratio					
pH 7.8	1.5 (±0.18)	3.5 (±0.66)	strobilation type at pH 7.8	*p* < 0.001	***
pH 8.1	2.1 (±0.28)	3.6 (±0.72)	strobilation type at pH 8.1	*p* < 0.001	***
			pH in polyp-strobilation	*p* < 0.001	***
			pH in planula-strobilation	*p* = 0.610	

NA, not applicable; data unavailable. Asterisk shows significant differences: **: *p* < 0.01, ***: *p* < 0.001.

## Data Availability

Data will be made available upon request.

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
