# Peer review of "Adaptation Strategy of the Planula Strobilation in Moon Jelly, Aurelia coerulea to Acidic Environments in Terms of Statolith Formation"

_animals, 2025, doi:10.3390/ani15131999_

Round 1
Reviewer 1 Report (Previous Reviewer 1)
Comments and Suggestions for Authors
The manuscript has improved significantly after revision. However, some minor changes are needed before it can be considered,
- Figure 2 provides a better quality picture
- Figure 3: Remove the arrow mark from the figure
- Table 2. Little consulting. If there is no effect, test statistics and P value for No. ephyrae analyzed and No. statoliths analyzed, it's better to add a dash or NA.
Author Response
General comment: The manuscript has improved significantly after revision. However, some minor changes are needed before it can be considered.
Response: Thank you very much for evaluating our manuscript and for your valuable comments on the manuscript. We carefully checked the comments and revised the manuscript.
Comments 1: Figure 2 provides a better quality picture
Response 1: We have replaced the images with higher-quality versions and rearranged them in the following order to improve clarity:
(a) Polyp-strobilation (poly-disc)
(b) Planula-strobilation (mono-disc).
Comments 2: Figure 3: Remove the arrow mark from the figure
Response 2: We have revised the figure as instructed and removed the arrow mark.
Comments 3: Table 2. Little consulting. If there is no effect, test statistics and P value for No. ephyrae analyzed and No. statoliths analyzed, it's better to add a dash or NA.
Response 3: Thank you very much for your valuable comment. As you suggested, we have revised Table 2 as Table 3
Reviewer 2 Report (New Reviewer)
Comments and Suggestions for Authors
General Comments:
The experimental design does not account for concurrent stressors (e.g., temperature rise, hypoxia), which limits ecological relevance. Future studies should integrate these variables. The morphological differences between strobilation pathways need further illustration. While “planula-strobilation” and “polyp-strobilation” are more easier to understand, aligning with established terms like “monodisc” and “polydisc” would increase acceptance. Tables and figures contain redundant information, which can be simplified.
Minor Comments:
Line 59: “witch”.
Line 103: Whether or not A. aurita in Germany has this phenomenon required verification.
Line 233: Table 1, it’s recommended to use “±” to express SE directly, and mark the unit clearly. The table can be appropriately beautified as well. Figures in bellow content can be added axes.
Line 404: “intensity”.
Author Response
General Comments: The experimental design does not account for concurrent stressors (e.g., temperature rise, hypoxia), which limits ecological relevance. Future studies should integrate these variables. The morphological differences between strobilation pathways need further illustration. While “planula-strobilation” and “polyp-strobilation” are more easier to understand, aligning with established terms like “monodisc” and “polydisc” would increase acceptance. Tables and figures contain redundant information, which can be simplified.
Response: Thank you for your careful review and constructive comments. We revised the manuscripts according to your comments. Our replies to your comments are as follows.
Comments 1: Line 59: “witch”.
Response 1: We have corrected the typo as suggested.
Comments 2: Line 103: Whether or not A. aurita in Germany has this phenomenon required verification.
Response 2: This phenomenon was indeed reported and sketched by Haeckel in 1881; however, no subsequent reports have been made, and whether it currently exists has not been verified.
Comments 3: Line 233: Table 1, it’s recommended to use “±” to express SE directly, and mark the unit clearly. The table can be appropriately beautified as well. Figures in bellow content can be added axes.
Response 3: Thank you for your suggestion. As recommended, we have revised the table to use the “±” symbol for standard error and clearly marked the units. We have also improved the table layout for clarity and added axes to the relevant figures.
Comments 4: Line 404: “intensity”.
Response 4: We have corrected the typo as suggested.
Reviewer 3 Report (New Reviewer)
Comments and Suggestions for Authors
Manuscript addresses ocean acidification, a critical climate change issue, and its biological impact on jellyfish reproduction, a subject of both ecological and evolutionary interest. Comparing two strobilation pathways (polyp-strobilation and planula-strobilation) under varying pH conditions is innovative and adds depth to our understanding of developmental plasticity in Aurelia coerulea.
Manuscript effectively reveals the experimental findings and the physiological and developmental adaptability of Aurelia coerulea under acidified ocean conditions. The comparison between polyp-strobilation and planula-strobilation is particularly informative, highlighting the functional plasticity of jellyfish reproduction and development in response to environmental stress.

Author Response
Comments 1:
Manuscript addresses ocean acidification, a critical climate change issue, and its biological impact on jellyfish reproduction, a subject of both ecological and evolutionary interest. Comparing two strobilation pathways (polyp-strobilation and planula-strobilation) under varying pH conditions is innovative and adds depth to our understanding of developmental plasticity in Aurelia coerulea.
Manuscript effectively reveals the experimental findings and the physiological and developmental adaptability of Aurelia coerulea under acidified ocean conditions. The comparison between polyp-strobilation and planula-strobilation is particularly informative, highlighting the functional plasticity of jellyfish reproduction and development in response to environmental stress.
Manuscript: Adaptation strategy of the planula strobilation in moon jelly, Aurelia coerulea to acidic environments in terms of statolith formation. The methodology is generally robust, and the results are meaningful and well supported. This study offers valuable insights into developmental plasticity under ocean acidification. However, clarity in replication, statistics, and figure presentation is needed to ensure transparency and reproducibility. Methods. Experimental Setup: The use of three pH conditions (6.8, 7.8, and 8.1) is appropriate and well justified—pH 6.8 mimics COâ‚‚ seep environments; pH 7.8 reflects 2100 RCP scenarios. pH control via COâ‚‚ injection and water chiller systems are well described. The consistency of environmental parameters is a major strength. The procedures for inducing polyp-strobilation and planula-strobilation are sound and well-documented, allowing for accurate comparisons between life history strategies. Measurement of Total Body Diameter (TBD), Central Disc Diameter (CDD), and statolith size/aspect ratio are appropriate and supported by prior literature. Use of ImageJ and established microscopy techniques is standard and appropriate for measuring statolith morphology.
Response:
Thank you very much for evaluating our manuscript and for your valuable comments on the manuscript. We carefully checked the comments and revised the manuscript.
Comments 1: The number of replicates (e.g., number of medusae or planulae per treatment) is unclear in some parts, especially regarding how many individuals were used per treatment over time. While two-way ANOVA and one-way ANOVA were applied, there is no mention of post hoc tests (e.g., Tukey’s HSD) to clarify which specific group differences are significant. ANOVA assumes homogeneity of variance, but there is no mention of other tests. This is a critical omission in statistical rigor.
Response 1: Thank you for your valuable and insightful comment. We have reconsidered the statistical analyses, and to ensure transparency, we have provided a more detailed description of both the methods and the results. Your suggestion significantly improved the clarity and rigor of our statistical reporting.
Comments 2: There is no mention of collection permits, ethical handling procedures, or marine sampling compliance—important even for invertebrates in many regions.
Response 2: Thank you for your valuable comment. In our case, jellyfish collection does not require a permit; therefore, this was not mentioned in the manuscript. However, ethical approval was obtained from our university, and this has now been stated in the “Institutional Review Board Statement” section located before the references.
Comment 3: The results clearly differentiate between ephyrae derived from the two reproductive pathways. Graphs, tables, and figures are effective and relevant. The finding that planula-strobilated ephyrae at pH 6.8 still form but lack statoliths is significant and novel. This could impact interpretations of developmental resilience in acidified oceans. The result that smaller but more numerous statoliths at pH 7.8 may preserve overall mass is insightful and may reflect an important physiological adaptation. Higher aspect ratios in planula-strobilated ephyrae support the hypothesis of early-stage crystallization. This provides a mechanistic insight into statolith biomineralization under stress.
Response 3: Thank you very much for your thoughtful and encouraging comments. We are pleased that you found our findings meaningful and novel. Your insights, particularly regarding developmental resilience in acidified oceans and physiological adaptations in statolith formation, are highly appreciated. We believe your comments greatly enhance the interpretation and significance of our results.
Round 2
Reviewer 3 Report (New Reviewer)
Comments and Suggestions for Authors
Manuscript is revised well, however some revision is needed by following comments.
1.While the number of ephyrae analyzed is mentioned (e.g., n = 10), it's unclear how many individuals per group or per tank were originally used, and whether biological or technical replicates were prioritized.
2.Please clarify how variability across tanks was controlled or accounted for.
3.Although robust ANOVA was applied correctly for heteroscedastic data, it would be helpful to briefly justify the use of trimmed means ANOVA in the methods section for clarity.
4.Some figure legends (e.g., Figures 6–10) are lacking detailed explanations about sample sizes, significance levels, and variability (e.g., error bars, p-values).
5.Ensure consistency in statistical annotation across figures and tables (e.g., * vs ** vs *** symbols).
6.The conclusion that planula-strobilation is less affected by acidification than polyp-strobilation contradicts the original hypothesis stated in the introduction. This should be acknowledged and discussed more clearly.
7.The term "biological minimum state" is conceptually interesting but could benefit from clearer definition and possibly referencing related ecological concepts.
8. pH values should consistently be written as “pH 6.8”, not “pH6.8”. provide space, check in figure captions as well.
9. Units like “µm²” and “mm” should be consistently spaced (e.g., “114.6 ± 31.8 µm²”).
10. Consider including more discussion on ecological/functional implications of altered statolith formation, possibly referencing movement studies or predator avoidance.
Author Response
General comment: Manuscript is revised well, however some revision is needed by following comments.
Response: Thank you very much for your positive feedback and for recognizing the improvements in our revised manuscript. We sincerely appreciate your additional comments and suggestions. In response, we have carefully addressed each point and made the necessary revisions accordingly. Please find our point-by-point responses below.
Comment 1: While the number of ephyrae analyzed is mentioned (e.g., n = 10), it's unclear how many individuals per group or per tank were originally used, and whether biological or technical replicates were prioritized.
Response 1: Thank you for pointing it out. The number of ephyrae analyzed was written as biological replicates. We revised the manuscripts according to your comments. Please check chapter 2: Materials and Methods.
Comment 2: Please clarify how variability across tanks was controlled or accounted for.
Response 2: In this study, each pH condition was represented by a single tank. Therefore, variability between tanks could not be statistically separated from the effects of pH. We acknowledge this as a limitation of the experimental design, and interpretations are made with this constraint in mind. We revised chapter 2: Materials and Methods to clarify it.
Comment 3: Although robust ANOVA was applied correctly for heteroscedastic data, it would be helpful to briefly justify the use of trimmed means ANOVA in the methods section for clarity.
Response 3: We have revised the manuscript in accordance with your comments. Please check chapter 2.6 Statistical analysis.
Comment 4: Some figure legends (e.g., Figures 6–10) are lacking detailed explanations about sample sizes, significance levels, and variability (e.g., error bars, p-values).
Response 4: We have added the requested details to the figure legends of Figures 6–10.
Comment 5: Ensure consistency in statistical annotation across figures and tables (e.g., * vs ** vs *** symbols).
Response 5: Thanks a lot for noticing this point. The manuscript has been modified based on your suggestions.
Comment 6: The conclusion that planula-strobilation is less affected by acidification than polyp-strobilation contradicts the original hypothesis stated in the introduction. This should be acknowledged and discussed more clearly.
Response 6: We modified in accordance with your opinion. Please check chapter 4: Discussion.
Comment 7: The term "biological minimum state" is conceptually interesting but could benefit from clearer definition and possibly referencing related ecological concepts.
Response 7: This is an excellent point, and we thank you for highlighting it. As per your pointing out, we have revised the relevant sections, chapter 4: Discussion.
Comment 8: pH values should consistently be written as “pH 6.8”, not “pH6.8”. provide space, check in figure captions as well.
Response 8: We are grateful for your careful confirmation. We revised manuscript and figures to unify notation.
Comment 9: Units like “µm²” and “mm” should be consistently spaced (e.g., “114.6 ± 31.8 µm²”).
Response 9: We are grateful for your careful confirmation. We revised manuscript and figures to unify notation.
Comment 10: Consider including more discussion on ecological/functional implications of altered statolith formation, possibly referencing movement studies or predator avoidance.
Response 10: Thank you for your thoughtful remarks. However, we believe it would not be appropriate to extend the discussion that far based on the current results. This study specifically examined how pH and strobilation types affect body morphology and statolith formation of ephyrae. Since we did not investigate predator presence or directly assess the relationship between statolith morphology and swimming ability, we feel that such ecological or functional interpretations would be speculative at this stage.
Orhers:
To improve the paper, we also made the following changes:
- Following the above, we also listed the CDD values. (2.2. Polyp-strobilated ephyrae and 3.2.3. Planula-strobilated ephyrae)
- We have described in more detail the ephyra formed at pH 6.8. (3.2. Polyp-strobilated ephyrae and 3.3.3. Planula-strobilated ephyrae)
- To make the notation more consistent, we have changed the order of the pH values in Table 1, Table 3, and Figure 9.
This manuscript is a resubmission of an earlier submission. The following is a list of the peer review reports and author responses from that submission.
Round 1
Reviewer 1 Report
Comments and Suggestions for Authors
The manuscript entitled Adaptation strategy of the planula strobilation in moon jelly, Aurelia coerulea to acidic environments with regard to statolith formation is very interesting and well-designed. A very important aspect, i.e., the effect of acidification on Aurelia coerulea, was investigated.
However, some issues need to be addressed before the paper can be considered for publication.
1. On what basis the pH was selected for the study
2. Why pH below 6.8 and above 8.1 were not used to see the effect on ephyrae of Aurelia coerulea
3. More information on growth conditions and the ecological importance of Aurelia coerulea needs to be highlighted in the introduction and discussion section.
4. Provide a reference to support these claims
Living in a healthy state under acidic oceans will be difficult for Aurelia medusa.
Planula-strobilation may be easily affected by acidic 94 environments.
5. Very old references are used in the paper. It's better to refer recent papers and compare the results with them. Accordingly, improve the discussion and introduction section.
Reviewer 2 Report
Comments and Suggestions for Authors
There are several key inaccuracies in the Introduction. You end paragraph 1 by saying "little is known about the impact of ocean acidification on Aurelia" (line 62), but several studies have been published on this topic, so that is not accurate. On line 72, you speak about settlement of Pelagia and Periphylla planulae on substrates, fully ignoring the fact that both genera are holoplanktonic! Lines 80-83 are not clear. On line 87 you speak about the use of sulfate-free seawater to rear jellyfish, yet your paper focuses on pH effects - what is the point? Lines 92-100 again, not clear. Lines 114, 133 - what species of jellyfish were collected? Line 120 - what were polyps fed? what was the light/dark cycle set for? Line 124 - why did you raise temperature to 23 C to rear the polyps after a regime of 11 C? line 127 - what colonies are you talking about?? These are medusae - not bacteria! line 128 - induced strobilation by dropping the temperature: was this your method, or was it published by someone else, and if so - please cite them! lines 137-141 - not clear... Overall, a lot of the text was difficult to follow and not well organized. By focusing on the comparison between polyp and planula strobilation, it was not clear what you were trying to achieve, especially when you presented data on ephyra morphology and behavior, statolith size and shape, the effect of pH on all of these variables, the effect of different temperatures... because you presented so many variables and effects, it became very difficult to keep track of what was causing what. Even the tables you presented were very difficult to read and understand, making it even harder to make sense of what you found and were trying to point out. By the time I reached the Discussion, I hoped to gain some understanding, but the Discussion was just as difficult to follow as the Results.
Comments on the Quality of English LanguageThe English needs to be edited by English speaking editor.